# Performance and Safety of EUS Ablation Techniques for Pancreatic Cystic Lesions: A Systematic Review and Meta-Analysis

**DOI:** 10.3390/cancers15092627

**Published:** 2023-05-05

**Authors:** Apostolis Papaefthymiou, Gavin J. Johnson, Marcello Maida, Paraskevas Gkolfakis, Daryl Ramai, Antonio Facciorusso, Marianna Arvanitakis, Alexander Ney, Giuseppe K. Fusai, Adrian Saftoiu, Daniela Tabacelia, Simon Phillpotts, Michael H. Chapman, George J. Webster, Stephen P. Pereira

**Affiliations:** 1Pancreaticobiliary Medicine Unit, University College London Hospitals (UCLH), London NW1 2BU, UK; a.papaefthymiou@nhs.net (A.P.); gavin.johnson2@nhs.net (G.J.J.); simon.phillpotts@nhs.net (S.P.); michael.chapman1@nhs.net (M.H.C.); george.webster1@nhs.net (G.J.W.); 2Gastroenterology and Endoscopy Unit, S. Elia-Raimondi Hospital, 93100 Caltanissetta, Italy; marcello.maida@hotmail.it; 3Department of Gastroenterology, General Hospital of Nea Ionia “Konstantopoulio-Patision”, 14233 Athens, Greece; pgolfakis@gmail.com; 4Department of Gastroenterology, Hepatopancreatology and Digestive Oncology, Erasme University Hospital, Université Libre de Bruxelles, 1070 Brussels, Belgium; marianna.arvanitaki@hubruxelles.be; 5Gastroenterology and Hepatology, University of Utah Health, Salt Lake City, UT 84132, USA; daryl.ramai@hsc.utah.edu; 6Gastroenterology Unit, Department of Surgical and Medical Sciences, University of Foggia, 37920 Foggia, Italy; antonio.facciorusso@virgilio.it; 7Institute for Liver and Digestive Health, University College London, London NW3 2PF, UK; alexander.ney.15@ucl.ac.uk; 8Department of HPB Surgery and Liver Transplantation, Royal Free Hospital, London NW3 2QG, UK; g.fusai@nhs.net; 9Research Center of Gastroenterology and Hepatology Craiova, University of Medicine and Pharmacy “Carol Davila”, 4192910 Bucharest, Romania; adriansaftoiu@gmail.com (A.S.); daniela.tabacelia@gmail.com (D.T.)

**Keywords:** pancreatic cyst, EUS, ablation, ethanol, RFA, paclitaxel, lauromacrogol

## Abstract

**Simple Summary:**

Pancreatic cystic lesions are diagnosed with an increasing frequency, thus comprising a significant routine condition in clinical practice. In addition to the current approaches, which include surgery and surveillance, endoscopic ultrasound (EUS) provides the potential of an additional therapeutic tool. This review collected the existing literature regarding EUS-guided ablation techniques for pancreatic cystic lesions and assessed its efficacy and safety. The cumulative effect in treating pancreatic cysts was 44% (95%CI: 31–57), with the highest rate achieved when a combination of ethanol and paclitaxel was injected into the cysts (70%; 95%CI: 64–76). Considering safety, most adverse events were mild and occurred after ethanol injection. EUS-guided pancreatic cyst ablation seems to be an acceptable and safe procedure, with promising results in appropriately selected patients.

**Abstract:**

Background: Pancreatic cystic lesions (PCL) represent an increasingly diagnosed condition with significant burden to patients’ lives and medical resources. Endoscopic ultrasound (EUS) ablation techniques have been utilized to treat focal pancreatic lesions. This systematic review with meta-analysis aims to assess the efficacy of EUS ablation on PCL in terms of complete or partial response and safety. Methods: A systematic search in Medline, Cochrane and Scopus databases was performed in April 2023 for studies assessing the performance of the various EUS ablation techniques. The primary outcome was complete cyst resolution, defined as cyst disappearance in follow-up imaging. Secondary outcomes included partial resolution (reduction in PCL size), and adverse events rate. A subgroup analysis was planned to evaluate the impact of the available ablation techniques (ethanol, ethanol/paclitaxel, radiofrequency ablation (RFA), and lauromacrogol) on the results. Meta-analyses using a random effects model were conducted and the results were reported as percentages with 95% confidence intervals (95%CI). Results: Fifteen studies (840 patients) were eligible for analysis. Complete cyst resolution after EUS ablation was achieved in 44% of cases (95%CI: 31–57; 352/767; I^2^ = 93.7%), and the respective partial response rate was 30% (95%CI: 20–39; 206/767; I^2^ = 86.1%). Adverse events were recorded in 14% (95%CI: 8–20; 164/840; I^2^ = 87.2%) of cases, rated as mild in 10% (95%CI: 5–15; 128/840; I^2^ = 86.7%), and severe in 4% (95%CI: 3–5; 36/840; I^2^ = 0%). The subgroup analysis for the primary outcome revealed rates of 70% (95%CI: 64–76; I^2^ = 42.3%) for ethanol/paclitaxel, 44% (95%CI: 33–54; I^2^= 0%) for lauromacrogol, 32% (95%CI: 27–36; I^2^ = 88.4%) for ethanol, and 13% (95%CI: 4–22; I^2^ = 95.8%) for RFA. Considering adverse events, the ethanol-based subgroup rated the highest percentage (16%; 95%CI: 13–20; I^2^ = 91.0%). Conclusion: EUS ablation of pancreatic cysts provides acceptable rates of complete resolution and a low incidence of severe adverse events, with chemoablative agents yielding higher performance rates.

## 1. Introduction

Pancreatic cystic lesions (PCL) have been increasingly diagnosed over the last few years due to the increased number of abdominal scans being undertaken, and the improved resolution of cross-sectional imaging. PCLs are diagnosed incidentally in 2–20% of computed tomography (CT) or magnetic resonance imaging (MRI) scans performed for unrelated non-pancreatic indications [1,2]. Due to the malignant potential of some PCLs, a specialist decision is then required on the optimal management strategy for patients found to have a PCL, with the majority entering a surveillance program [3,4,5,6,7]. Pancreatic ductal adenocarcinoma (PDAC) and pancreatic neuroendocrine neoplasms (pNEN) rarely present as cystic lesions [8]. Intraductal papillary mucinous neoplasms (IPMN) are the most common types of PCL with malignancy potential, and with mucinous cystic neoplasms (MCN) and solid pseudopapillary neoplasms (SPN) also inferring a risk of malignancy. Pseudocysts, serous cystadenomas (SCN) and other rare cysts are essentially benign, however they cannot always be accurately classified with the available modalities, particularly when of a small size [9].

Position statements and numerous guidelines have been developed to support clinicians in the diagnosis and management of PCLs, with the two principal outcomes being either surgical resection or ongoing surveillance, depending on various clinical, laboratory and imaging criteria [3,4,5,6]. Although surveillance is often non-invasive with axial imaging, and therefore safe, there are downsides for patients. Patients live with concerns regarding the cancer risk and the anxiety accompanying follow-up procedures, and significant somatization, depression, reduced health perception and functionality is not uncommon compared with those managed surgically [10,11,12]. Moreover, economic evaluation studies of the most broadly adopted guidelines have questioned the value of current approaches in terms of cost-effectiveness, whereas surgical overtreatment provides comparable results due to the increased iatrogenic morbidity and mortality [13,14]. This is relevant to the high level of surgical complexity, depending on the type of lesion and location, as these patients undergo radical excisions with perioperative and long-term impact on their health.

In this regard, the minimally invasive nature of therapeutic endoscopic ultrasound (EUS) ablation could be an alternative solution to surveillance or surgery. Ethanol and radiofrequency ablation (RFA) have been used for decades as complementary techniques to surgical or systematic therapies for the treatment or palliation of patients with cancer, including PDAC [15,16]. EUS-guided RFA with dedicated devices or injection therapy via conventional needles have been studied in adrenal masses and pNENs, with promising results in terms of technical success, symptomatic relief, and even total cure [17,18,19]. Nevertheless, there is no clear consensus for the role of EUS ablation for PCL management, despite the existence of published studies, probably due to the absence of an overall estimation of its efficacy and safety. In 2019, an international position statement about PCL EUS ablation was published, based on weak evidence, supporting the use of ethanol ablation for unilocular or oligolocular mucinous or enlarging PCLs in patients choosing to avoid surgery, or in poor surgical candidates with a reasonable life expectancy [20].

This systematic review with meta-analysis aims to accumulate the data on therapeutic performance of EUS ablation of PCL in terms of complete cyst resolution, partial response, and adverse events, and to identify differences between the available ablation techniques.

## 2. Materials and Methods

Our research was based on a detailed study protocol, which was registered in the prospective platform for systematic reviews (PROSPERO registration number: CRD42023412014). The structure and methodology of our study was based on the Preferred Reporting Items for Systematic Reviews and Meta-Analyses (PRISMA) checklist (Appendix A) [21].

### 2.1. Inclusion and Exclusion Criteria

The validated PICO framework was the basis of the primary question of this review including the assessment of EUS ablation in terms of complete cyst disappearance, partial response, and adverse events [22]. Case-series or cohorts evaluating this therapeutic approach were included in the final analysis when the following criteria were accomplished: (A) patients: adult patients (>18 years old), with pancreatic cyst, using (B) interventions: EUS ablation techniques with ethanol, RFA, or ethanol with paclitaxel or lauromacrogol; (C) comparators: given the design of existing studies in the literature, no comparisons were feasible; (D) outcomes: complete cyst resolution (defined as the complete disappearance of the cystic lesion after EUS ablation based on imaging (MRI, CT or EUS) during the follow up period); partial resolution (representing the reduction in cyst size after EUS ablation, without complete disappearance, as measured by the imaging modality used in the follow up scan); and the rate of adverse events related to the procedure. Case series with less than 10 patients, case reports, abstracts, and studies with incomplete data or inadequate follow-up (less than 48 h for post-procedure adverse events and less than one month for treatment success) were excluded from our analysis.

### 2.2. Search Strategy

An initial search was performed using PubMed/MEDLINE, Cochrane and Scopus databases, on 2 April 2023. The search algorithm included the following Boolean search terms: (“pancreatic cyst” OR “intraductal papillary mucinous neoplasm” OR “mucinous pancreatic cyst” OR serous pancreatic cyst”) AND (“EUS ablation” OR “EUS RFA” OR “ethanol ablation”). Additional relevant articles were hand-searched in the reference lists of the retrieved publications as well as by using the “similar article” function within PubMed. Unpublished works, abstracts, and oral or poster presentations were excluded. In cases of missing data, the first and/or the corresponding authors were contacted. Two investigators (AP, PG) independently selected articles of interest based on the aforementioned inclusion and exclusion criteria. In cases of multiple publications from the same study, only the most recent and complete article was included.

### 2.3. Data Abstraction and Quality Assessment

Data on study-, participant-, and intervention-related parameters were retrieved into a standardized form by two investigators (AP, DR) independently; discrepancies were resolved by consensus, referring to the original article, after consultation with a third reviewer (GJ). The quality of the included studies was assessed by two authors independently (DR, AF) using the validated Newcastle–Ottawa scale.

### 2.4. Outcomes

The primary outcome of our meta-analysis was complete cyst resolution after EUS ablation. The secondary outcomes included: (1) partial size reduction, and, (2) the rate of adverse events. Where available, adverse events were classified as early or late, depending on the definitions of the included studies. Mild adverse events included post-procedural pain, minor bleeding or fever, without requiring admission or further interventions; whereas severe included perforation, pancreatitis, severe infection, adjacent structures injury, significant hemorrhage requiring blood products and/or additional interventions, or unplanned hospital admission related to the procedure.

### 2.5. Statistical Analysis

Pooled proportions and 95% confidence intervals (CIs) were calculated by using the Der Simonian and Laird random-effects model that incorporates both between-study and within-study variation [23].

Heterogeneity between study-specific estimates was assessed using the inconsistency index (I^2^), and cut-off points of <30%, 30–59%, 60–75%, and >75% were considered to suggest low, moderate, substantial, and considerable heterogeneities, respectively [24].

A subgroup analysis was conducted to assess the potential different effect of the available ablation techniques (RFA, ethanol, ethanol and paclitaxel or lauromacrogol). All results between the subgroups were compared to investigate statistically significant differences. Publication bias was estimated after visual assessment of the funnel plot for the primary outcome [25].

For all analyses, a *p* value of <0.05 was considered statistically significant. The analyses were performed using R packages [26].

### 2.6. Quality of Evidence

The quality of the provided evidence was rated based on GRADE criteria [27].

## 3. Results

### 3.1. Characteristics of Included Studies

After applying the exclusion criteria, 15 studies [28,29,30,31,32,33,34,35,36,37,38,39,40,41,42] (840 patients) out of 1114 studies were eligible for inclusion. The selection PRISMA flowchart is illustrated in Figure 1, and Table 1 summarizes the main characteristics of the included studies. Most of the studies were developed in the U.S.A. [31,33,34,35,37,38], five in Korea [28,29,30,39,42], two in China [32,36], whereas the only European country which published data on EUS ablation for cysts was France (two studies) [40,41]. In terms of study design, two randomized controlled studies (RCTs) were identified [31,38], and nine were prospective cohorts [28,29,32,33,34,35,36,39,41]. The remaining four cohorts were retrospectively collected [30,37,40,42]. Eight studies [28,33,34,35,37,38,40,42] evaluated ethanol ablation, three studies evaluated its combination with paclitaxel [29,33,39], and one study evaluated a triple regimen of ethanol, paclitaxel and gemcitabine [31]. Of the four remaining studies, two used RFA [28,41], and two used lauromacrogol [32,36].

The age range of study participants was from 20 to 85 years, and the female to male ratio was 2:1. All studies followed up their patients for at least 12 months, with a maximum recorded surveillance time of 119 months after EUS ablation. In ten studies, the follow-up started 2–4 months after EUS ablation [28,29,30,32,33,36,37,38,39,40], and in five studies, surveillance started after 6 months [31,34,35,41,42]. In total, 73 patients were lost to follow-up, and, therefore, 767 (91.3%) were included in the analysis. The majority of PCL were located in the body/tail of the pancreas, whereas 325 (40.1%) were in the head of the pancreas. The range of mean cyst size was 19.4 to 50 mm, and considering PCL subtypes, 265 (31.5%) MCN were included, followed by 209 (24.9%) SCN, 188 (22.4%) side branch (SB)-IPMNs, and 44 pseudocysts (5.2%), whereas 132 (15.7%) PCL were indeterminate at the time of ablation. Mural nodules were identified in 21 cysts (2.5%).

### 3.2. Quality Assessment

All of the included studies were of good quality. More specifically, the two RCTs were graded with the maximum score, fulfilling all the required parameters of the NOS [31,38]. Apparently, the only drawback of the remaining cohort studies was the absence of comparator, thus yielding the optimal quality given their design (Appendix A).

### 3.3. Primary Outcome—Complete Cyst Resolution

Pancreatic cysts were completely resolved at a rate of 44% (95%CI: 31–57; 352/767; I^2^ = 93.7%) at least 12 months post EUS ablation (Figure 2).

### 3.4. Secondary Outcomes

Partial response with size reduction was recorded in 30% (95%CI: 20–39; 206/767; I^2^ = 86.1%) of cases (Appendix A).

### 3.5. Adverse Events

Procedure related adverse events were described in 14% (95%CI: 8–20; 164/840; I^2^ = 87.2%) of cases (Figure 3). Most of these were mild (10%; 95%CI: 5–15; 128/840; I^2^ = 86.7%), whereas severe complications accounted for 4% (95%CI: 3–5; 36/840; I^2^ = 0%), with null heterogeneity (Appendix A). The most prevalent adverse event was post-procedural pain in 89 (10.6%) patients, followed by pancreatitis (43 patients, 5.1%) which was mainly mild. One patient had perforation, and four had procedure-related bleeding. All of the recorded adverse events were early, reported in 2–7 days post-procedure, except for two pseudocysts, two abscesses, one portal vein thrombosis, one splenic vein obliteration, one duodenal stricture, and two main pancreatic duct strictures recorded 14–30 days later [28,39,42]. One death was recorded 41 months after ablation, due to adenocarcinoma development on the site of the ablated cyst [34].

### 3.6. Subgroup Analysis

Subgroup analysis for the primary outcome resulted in a rate of 32% (95%CI: 27–36; I^2^ = 88.4%) for ethanol, and 13% (95%CI: 4–22; I^2^ = 95.8%) for RFA, with their difference been statistically significant (*p* = 0.004). Interestingly, the combination of ethanol and paclitaxel yielded complete cyst disappearance in 70% of cases (95%CI: 64–76; I^2^ = 42.3%), and lauromacrogol in 44% of cases (95%CI: 33–54; I^2^= 0%) (Appendix A).

Partial cyst resolution did not differ between ethanol (36%; 95%CI: 31–40; I^2^ = 91.0%) and RFA (38%; 95%CI: 21–54; I^2^ = 80.1%) subgroups (*p* = 0.75) (Appendix A). Ethanol/paclitaxel ablation achieved partial response in 18% (95%CI: 12–23; I^2^ = 28.3%) and lauromacrogol in 29% (95%CI: 19–38; I^2^ = 0%) of cases, with null heterogeneity.

EUS ablation with ethanol demonstrated the highest risk of adverse events (16%; 95%CI: 13–20; I^2^: 91.0%), with marginally non-significant difference from RFA (7%; 95%CI: 0–12; I^2^ = 0%; *p* = 0.08). The complication rates for ethanol/paclitaxel and lauromacrogol were 9% (95%CI: 5–13; I^2^ = 0%) and 5% (95%CI: 1–10; I^2^ = 0%), respectively. All techniques apart from ethanol ablation achieved null heterogeneity (Appendix A).

### 3.7. Quality of Evidence

Given that the majority of the included studies were observational, the quality of evidence was rated as low. No reasons for further downgrading were recognized. Therefore, based on the meta-analysis, the low quality of evidence supported the comparisons among the presented modalities.

### 3.8. Publication Bias

The funnel plot considering the primary outcome is presented in Appendix A, and the noticed symmetry indicates the absence of publication bias.

## 4. Discussion

This review with meta-analysis is the first based on good quality studies that assesses the pooled performance of all available EUS-guided ablative techniques to treat pancreatic cysts. The cumulative rate of complete PCL resolution, which was the main therapeutic aim, was 44% (95%CI: 31–57), though accompanied with high heterogeneity, ranging between 4% and 85% in the individual studies. The highest success rate in complete PCL treatment was evidenced from the two studies of combined ethanol and paclitaxel use, achieving a resolution rate of 70% (95%CI: 64–76), followed by lauromacrogol which homogenously resolved 44% (95%CI: 33–54) of PCL. Although the success rate of ethanol used in isolation was lower (32%; 95%CI: 27–36), it was based on more studies, and remained significantly higher than RFA (13%; 95%CI: 4–22). Interestingly, the rate of the partial response for all ablation techniques was inferior (30%; 95%CI: 20–39), with similar rates among ethanol, RFA and lauromacrogol, and lower for the ethanol/paclitaxel group, probably reflecting its higher rate of complete resolution with this technique.

Regarding safety, the overall frequency of adverse events was 14% (95%CI: 8–20), albeit with significant heterogeneity, as this percentage ranged between 2% and 41% among included studies. A major source of this variability was the ablation technique. The limited number of studies assessing RFA, ethanol/paclitaxel and lauromacrogol, resulted in null heterogeneity, with minimal overall rates of complications (5–9%). On the other hand, ethanol ablation was reported to have an incidence of adverse events which varied quite markedly between studies, with a mean percentage of 16% (95%CI: 13–20). Moreover, from the 36 recorded severe adverse events, 31 (86.1%) occurred after ethanol or ethanol/paclitaxel ablation. In addition to the reported complications, chemoablation was reported as associated with the development of new cysts, similar to pseudocysts, with specific radiological and histological characteristics [43]. Choi et al. [42] in their patient cohort treated with ethanol, indicated that SB-IPMNs, multilocular cysts, suspected ethanol leakage during the procedure, and increased cystic fluid viscosity were independently associated with pancreatitis, however their conclusions were not confirmed by other reports. Two cases of adjacent structures injury were recorded in our study, including a case of splenic vein thrombosis after ethanol ablation, and one of jejunal perforation after RFA [39,41]. Importantly, no deaths were reported as direct complications of ablation and, despite the relatively high rate of overall complications, the majority of adverse events (10%; 5–15%) were mild, and included pain, self-limited intra-cystic bleeding, or fever. The recommended use of antibiotics during PCL puncture is likely to have prevented major infection [44]. Regarding the time of the recorded adverse events, all studies followed up patients for at least 48 h post-procedure. The vast majority of adverse events were early, manifesting during the first week after treatment. Complications two weeks after ablation were considered as late, including two pseudocysts, two abscesses, one portal vein and one splenic vein thrombosis, one duodenal stricture and two main pancreatic duct strictures occurring during the first month, with the overall follow-up being at least 12 months per study. However, in one study using ethanol, despite the complete PCL resolution after ablation, one patient developed PDAC 3.5 years later and died, thus, raising the concern of long-term recurrence rates and potential effects on the normal parenchyma, requiring prospective studies with strict follow-up protocols [34].

A key question about EUS ablation for PCL is the selection of the most appropriate modality. Ethanol is the most thoroughly studied ablative material compared with the alternatives, though with variable outcomes and significant rates of adverse events. An apparent limitation of all injection-based techniques is the existence of lobulated cysts, where cyst aspiration and agent injection into all aspects of the lesion are not feasible. Specifically, the identification of more than six locules or a mural nodule has been implicated in lower success rates of chemoablation [45]. In these cases, thermal ablation using EUS-RFA could potentially represent a reasonable alternative. Although our results indicate a limited value for RFA, they are based on only two studies. In a prospective study, Barthet et al. [41] recorded the third highest rate of complete CP resolution (65%) among available techniques, followed by a similarly high rate in a small case series study [46]. On the other hand, Oh et al. [29] presented the lowest EUS-RFA success rate (4%). Interestingly, all PCLs included in the first study were MCN or SB-IPMNs, whereas Oh et al. [29] recruited patients with SCN, thus indicating that in cases where it is clinically significant, RFA may have a high success rate. Despite this broad heterogeneity, the scarcity of studies on EUS-RFA for PCL and its high success rate on other pancreatic lesions underlines the necessity for further study, at least for PCL with specific characteristics [47,48]. Moreover, as with ethanol and paclitaxel, the combination of ablative techniques could be beneficial in terms of complete resolution. Moyer et al. [31] used ethanol, paclitaxel and gemcitabine to treat PCL, achieving a high rate (61%) of complete response. Interestingly, they also revealed similar rates (67%) using an ethanol-sparing comparator, indicating that this approach can provide equally high therapeutic results protecting from the ethanol-related adverse events [31]. RFA combined with lauromacrogol injection has also been assessed in isolated reports, resulting in cyst disappearance, and implying that this may be a promising approach worthy of further assessment [49].

The presented results should be interpreted in light of the position EUS ablation sits in the PCL management algorithm. To date, international guidelines do not yet advocate EUS-guided ablation for PCL, and so most candidates for EUS ablation are patients who decline surgical treatment, or those who are unfit for an operation [20,45]. The expansion of this indication into all patients with definite risk factors for malignancy on the basis of PCL seems unlikely, based on our pooled rates of complete resolution. However, it could be a reasonable alternative for appropriate cystic lesions in patients unwilling to undergo annual follow-up, or with persistent impact of surveillance on their quality of life. Cases with indefinite cysts of uncertain significance may benefit from an ablation approach compared with surveillance. In the presented studies, initial assessment of treatment response started 2–6 months after the procedure, depending on the individual study protocol, and all patients were observed for at least 12 months for complete resolution or recurrence. Long-term studies, comparing EUS ablation to surgery and surveillance in terms of efficacy, recurrence, adverse events, cost-effectiveness and quality of life, will be needed before recommending application of PCL ablation more broadly. Choi et al. [50] in a propensity score matching analysis—using a subset of patients included in our meta-analysis—compared ethanol ablation to the natural course of PCL. Although the mortality between the two groups was not different, the frequency of surgery was significantly less in the EUS ablation group (4.8%) compared with the control (26.2%), thus improving patients’ quality of life and reducing health-care expenditure [50]. A useful tool would also be the establishment of predictive factors of response after EUS ablation. Unilocular morphology and smaller size have been identified as potential predictors of cyst resolution, but further variables, including modality-specific data, need to be identified [39]. Although the benefit of complete cyst disappearance is clear, the significance of partial response has not been demonstrated, and results in the need for ongoing cyst surveillance. Despite this, all the studies used partial response as a secondary outcome, with various percentages of cyst volume reduction as a definition, but whether this reduces the risk of malignant transformation or the subsequent need for surgery is not clear.

This study has some limitations. The most significant one was the design of some of the included studies. Moreover, all studies, except for the two RCTs, were observational, thus impacting on the overall quality of the retrieved data. This was reflected in our GRADE assessment, where the summary of evidence was classified as having low quality. Moreover, the absence of comparative studies among different ablation techniques did not facilitate direct comparisons regarding their performance. However, given the current guidelines and the low number of patients needing intervention, it is difficult to build studies comparing EUS ablation techniques or their results compared with surgery. The number of studies per modality was limited, except for those assessing ethanol, thus not allowing the generalization of subgroup meta-analyses results. Furthermore, the scarcity of data on technical success did not allow relative analysis to identify difficulties and obstacles from an endoscopic point of view, although the procedure is likely to be relatively straight-forward for experienced endosonographers. In most studies, the patients were offered more than one session of EUS ablation. However, the association between sequential ablations and efficacy/safety could not be evaluated based on the provided data. Finally, the lack of uniform protocols to assess variables that could affect the results was a significant barrier against identifying confounders and predictors associated with the outcomes. This is reflected in the heterogeneity, which remained mainly high, despite subgroup analysis. The different types of ablated cysts and the relatively broad range of their size are obvious factors resulting in heterogeneity; however, their assessment was not feasible due to the way of results presentation in the individual studies.

## 5. Conclusions

Pancreatic cysts are a common finding and lead to patient anxiety. Based on the presented results, EUS ablation may comprise an acceptable approach with a low incidence of severe adverse events. The most efficacious ablation technique, and for what type of lesion, needs further evaluation, before incorporating EUS ablation into defined PCL management guidelines. The evolution of therapeutic endosonography is now established as a vital part of the management of pancreatic diseases, but it remains to be seen if this is set to include the definitive treatment of PCLs.

## Figures and Tables

**Figure 1 cancers-15-02627-f001:**
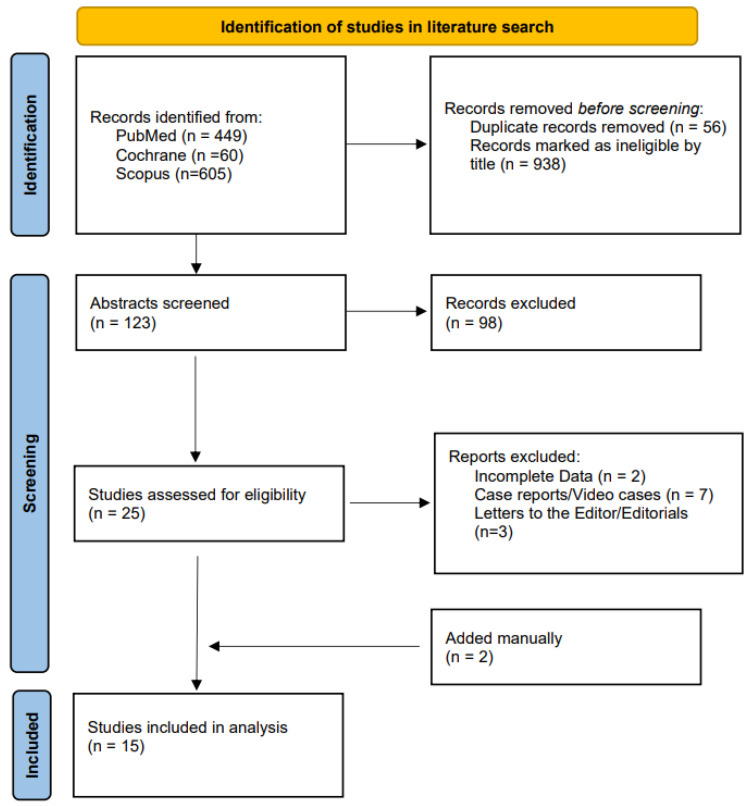
Study flowchart.

**Figure 2 cancers-15-02627-f002:**
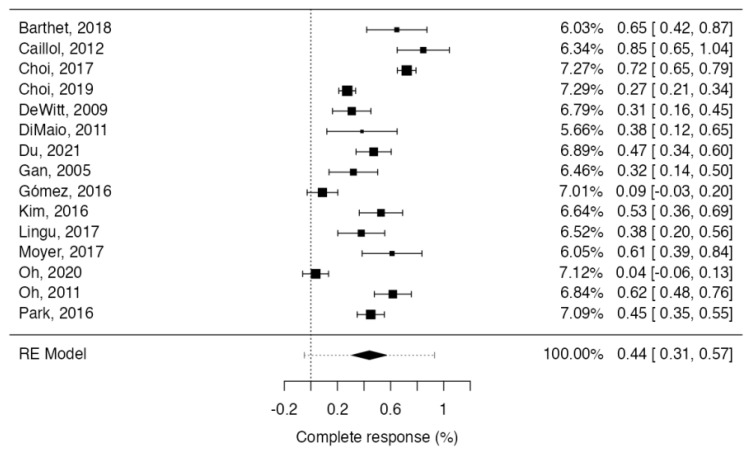
Forest plot reporting pooled results of the meta-analysis concerning complete cyst resolution [28,29,30,31,32,33,34,35,36,37,38,39,40,41,42].

**Figure 3 cancers-15-02627-f003:**
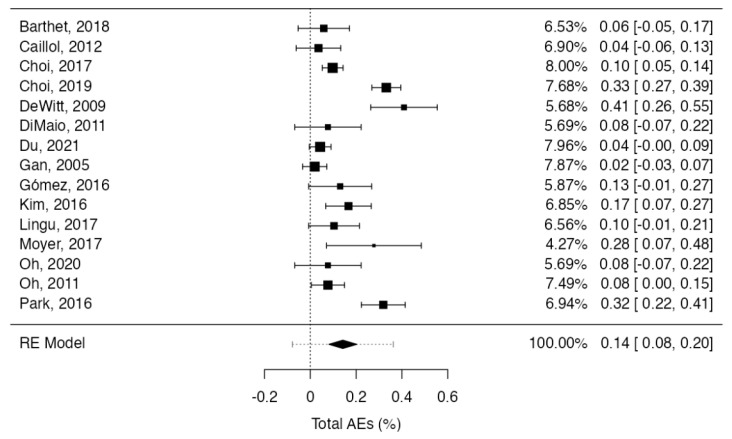
Forest plot reporting pooled results of the meta-analysis concerning the adverse events rate. [28,29,30,31,32,33,34,35,36,37,38,39,40,41,42].

**Table 1 cancers-15-02627-t001:** Main characteristics of included studies.

Study	Year	Design		Country	Recruitment Period	Number of Patients	Mean Age	Gender (% Female)	Mean Diameter (Range) mm	SB-IPMN (%)	MCN (%)	SCN (%)	Pseudocyst (%)	Indeterminate (%)	Head (%)	Body (%)	Tail (%)	Maximum Follow Up (Months)
Barthet et al.	2018	Prospective cohort	Multicenter	France	2 years	17	65.7 (range 65–83)	10 (59%)	28.0 (9.0–60.0)	16 (94%)	1 (6%)	0	0	0	10 (59%)	4 (24%)	3 (18%)	12
Caillol et al.	2012	Retrospective cohort	Two-center	France	2001–2010	13	68.5 (range 49–81)	6 (46%)	23.9 (11.0–50.0)	0	13 (100%)	0	0	0	9 (64%)	3 (21%)	2 (14%)	118
Choi et al.	2017	Prospective cohort	Single center	Korea	2005–2015	158	51 (IQR 20–85)	115 (70%)	Median 32.0 (IQR 26.0–41.0)	11 (7%)	71 (43%)	16 (10%)	3 (2%)	63 (38%)	42 (26%)	86 (52%)	36 (22%)	NR
Choi et al.	2019	Retrospective cohort	Single center	Korea	2006–2018	214	55.61 (SD ± 14.7)	140 (65%)	32.2 (SD ± 9.6)	63 (29%)	57 (27%)	69 (32%)	25 (12%)	0	89 (42%)	69 (32%)	56 (26%)	NR
DeWitt et al.	2009	Randomized trial, double-blind	Multicenter	USA	2004–2007	39	69.1 (SD ± 13.2)	27 (64%)	20.5 (10.0–40.0)	17 (41%)	17 (41%)	5 (12%)	3 (7%)	0	18 (43%)	16 (38%)	8 (19%)	39
DiMaio et al.	2011	Retrospective cohort	Single center	USA	2001–2008	21	70 (NR)	9 (69%)	20.1 (SD ± 7.1)	13 (100%)	0	0	0	0	8 (62%)	4 (30%)	1 (8%)	18
Du et al.	2021	Prospective cohort	Multicenter	China	2015–2020	70	50.3 (SD ± 14.2)	50 (71%)	32.0 (9.0–110.0)	0	27 (39%)	34 (49%)	0	9 (13%)	37 (53%)	23 (47%)		55
Gan et al.	2005	Pilot study	Single center	USA	2001–2003	25	64.5 (NR)	20 (80%)	19.4 (6.0–37.0)	3 (12%)	14 (56%)	3 (12%)	1 (4%)	1 (4%)	8 (32%)	8 (32%)	9 (36%)	12
Gómez et al.	2016	Prospective cohort	Single center	USA	2004–2014	23	70 (range 53–86)	10 (44%)	27.5 (14.9–49.3)	15 (65%)	4 (17%)	4 (17%)	0	0	15 (65%)	6 (26%)	2 (9%)	82
Kim et al.	2016	Prospective cohort	Single center	USA	2004–2015	36	69.1 (SD ± 12.2)	24 (67%)	25.8 (SD ± 8.7)	14 (39%)	16 (44%)	5 (14%)	1 (3%)	0	14 (39%)	22 (61%)		119
Linghu et al.	2017	Prospective cohort	Single center	China	2015–2016	29	56 (SD ± 15)	20 (68%)	30.6 (SD ± 15.0)	0	15 (52%)	12 (41%)	0	2 (7%)	NR	NR	NR	15
Moyer et al.	2017	Randomized trial, double-blind	Single center	USA	2011–2016	39	NR	23 (59%)	25 (15.5–42.0)	27 (69%)	9 (23%)	0	0	3 (8%)	19 (49%)	19 (49%)	1 (3%)	12
Oh et al.	2020	Retrospective cohort	Single center	Korea	2018–2019	13	60 (IQR 50.5–70)	5 (39%)	Median 50 (IQR 34.0–52.5)	0	0	13 (100%)	0	0	5 (39%)	8 (62%)		15
Oh et al.	2011	Prospective cohort	Single center	Korea	2005–2009	52	49.5 (range 22–81)	34 (65%)	31.8 (17.0–68.0)	0	9 (17%)	15 (29%)	2 (4%)	26 (50%)	16 (31%)	17 (33%)	19 (37%)	44
Park et al.	2016	Prospective cohort	Single center	Korea	NR	91	58 (range 28–83)	67 (74%)	30.0 (20.0–50.0)	9 (10%)	12 (13%)	33 (36%)	9 (10%)	28 (31%)	35 (38%)	32 (35%)	24 (26%)	117

IQR = Interquartile Range. SD = Standard Deviation. IPMN = Intraductal Papillary Mucinous Neoplasm. MCN = Mucinous Cystic Neoplasm. SCN = Serous Cystic Neoplasm. SB = Side Branch. NR = Not Reported.

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
