# Peer review of "Performance and Safety of EUS Ablation Techniques for Pancreatic Cystic Lesions: A Systematic Review and Meta-Analysis"

_cancers, 2023, doi:10.3390/cancers15092627_

Round 1
Reviewer 1 Report
Dear Authors,
this is a well written manuscript. The results are presented clearly, with appropriate discussion and conclusions drawn.
Although, similar works have already been published, this is the only known manuscript based on good quality studies and on pooled analysis of all EUS guided ablative techniques available to treat pancreatic cystic lesions.
I have one question regarding the definitions of outcomes in materials and methods section. I think you should specify the specific follow-up time-point at which the complete resolution and partial size reduction of pancreatic cystic lesions were assessed. In addition, the type of adverse events assessed with respect to the timing after EUS ablation is unclear (early and/or late).
Author Response
Answers to Reviewer #1
This is a well written manuscript. The results are presented clearly, with appropriate discussion and conclusions drawn.
Although, similar works have already been published, this is the only known manuscript based on good quality studies and on pooled analysis of all EUS guided ablative techniques available to treat pancreatic cystic lesions.
We appreciate Reviewer’s #1 encouraging comments.
I have one question regarding the definitions of outcomes in materials and methods section. I think you should specify the specific follow-up time-point at which the complete resolution and partial size reduction of pancreatic cystic lesions were assessed. In addition, the type of adverse events assessed with respect to the timing after EUS ablation is unclear (early and/or late).
For the majority of the included studies (10/15) the initial follow-up to assess response to EUS-ablation was performed 2-4 months after the procedure and continued at various intervals until a maximum of more than 12 months. The remaining 1/3 of studies used a 6-months' time-point to evaluate the initial response.
In 12 out of 15 studies, patients experienced early adverse events (in the first 2-7 days post-ablation). However, in the remaining 3 studies, 9 patients experienced late adverse events (14-30 days).
We have incorporated these data in the revised manuscript (pages 3, 4, 5, 8 of the revised manuscript)
Reviewer 2 Report
First at all, I wish to thank the authors for their nice work aiming to provide a rationale for the use of an appropriate EUS-guided therapeutic approach in the setting of these worrisome lesions, both for patients and clinicians.
I reccomemd this paper to be published, although before that I would make some commentaries than have been raised after review the manuscript.
1) Would it not be more advisable to choose the analysis of safety of the procedure as the main objective, followed by the rate of complete resolution?
2) I have missed in the discussion part a explanation about the lenght of the follow-up, both in the analysis of adverse events and, especially in the rate of complee and partial responses. If we choose any EUS-guided technique, it would be advisable to take into account the follow-up of each procedure. ALthough the follow-up in showed in table 1, I believe it would more clear for the reader and the clinician to have a vision of the time that these patients have been followed. Especially, when in two studies this data is not given.
Author Response
Answers to Reviewer #2
First at all, I wish to thank the authors for their nice work aiming to provide a rationale for the use of an appropriate EUS-guided therapeutic approach in the setting of these worrisome lesions, both for patients and clinicians. I recommend this paper to be published, although before that I would make some commentaries than have been raised after review of the manuscript.
We would like to thank Reviewer #2 for their positive statement about our study
Would it not be more advisable to choose the analysis of safety of the procedure as the main objective, followed by the rate of complete resolution?
We acknowledge Reviewer’s approach, as EUS-ablation of pancreatic cysts is not a widely established and studied technique, thus creating concerns about safety. However, our meta-analysis has been based on a predefined protocol, submitted to PROSPERO, with the primary outcome being considered as complete cyst resolution, following the rationale of the already published original studies, determining study selection. Any deviation from this structure would be outside of the protocol. Moreover, in our view the presentation of adverse events in the manuscript is detailed and clear, even as a secondary outcome, and we have now enriched this section with further information. Nevertheless, if Reviewer #2 and the Editor feel that it is necessary, we are prepared to restructure the outcomes.
I have missed in the discussion part an explanation about the length of the follow-up, both in the analysis of adverse events and, especially in the rate of complete and partial responses. If we choose any EUS-guided technique, it would be advisable to take into account the follow-up of each procedure. Although the follow-up in showed in table 1, I believe it would more clear for the reader and the clinician to have a vision of the time that these patients have been followed. Especially, when in two studies this data is not given.
We appreciate Reviewer’s comment and have modified the manuscript to include more details about patients’ follow up. (pages 3, 5, 8, 10, 11 of the revised manuscript)